# Prediction of Vaccine Response and Development of a Personalized Anti-SARS-CoV-2 Vaccination Strategy in Kidney Transplant Recipients: Results from a Large Single-Center Study

**DOI:** 10.3390/jpm12071107

**Published:** 2022-07-05

**Authors:** Ilies Benotmane, Gabriela Gautier-Vargas, Noëlle Cognard, Jérôme Olagne, Françoise Heibel, Laura Braun-Parvez, Jonas Martzloff, Peggy Perrin, Romain Pszczolinski, Bruno Moulin, Samira Fafi-Kremer, Sophie Caillard

**Affiliations:** 1Department of Nephrology, Dialysis and Transplantation, Strasbourg University Hospital, 1 Place de l’Hopital, BP 426, 67091 Strasbourg, France; gabriela.gautier-vargas@chru-strasbourg.fr (G.G.-V.); noelle.cognard@chru-strasbourg.fr (N.C.); jerome.olagne@chru-strasbourg.fr (J.O.); francoise.heibel@chru-strasbourg.fr (F.H.); laura.braun-parvez@chru-strasbourg.fr (L.B.-P.); jonas.martzloff@chru-strasbourg.fr (J.M.); peggy.perrin@chru-strasbourg.fr (P.P.); romain.pszczolinski@chru-strasbourg.fr (R.P.); bruno.moulin@chru-strasbourg.fr (B.M.); sophie.caillard@chru-strasbourg.fr (S.C.); 2Fédération de Médecine Translationnelle (FMTS), 67000 Strasbourg, France; samira.fafi-kremer@unistra.fr; 3Department of Virology, Strasbourg University Hospital, BP 426, 67091 Strasbourg, France

**Keywords:** kidney transplant recipients, COVID-19, anti-SARS-CoV-2 mRNA vaccine, serological response, prediction

## Abstract

Kidney transplant recipients (KTRs) displays marked inter-individual variations in magnitude of immune responses to anti-SARS-CoV-2 vaccination. The aim of this large single-center study was to identify the predictive factors for serological response to the mRNA-1273 vaccine in KTRs. We also devised a score to optimize prediction with the goal of implementing a personalized vaccination strategy. The study population consisted of 564 KTRs who received at least two doses of the mRNA-1273 vaccine. Anti-RBD IgG titers were quantified one month after each vaccine dose and until six months thereafter. A third dose vaccine was given when the antibody titer after the second dose was <143 BAU/mL. A score to optimize prediction of vaccine response was devised using the independent predictors identified in multivariate analysis. The seropositivity rate after the second dose was 46.6% and 22.2% of participants were classified as good responders (titers ≥ 143 BAU/mL). On analyzing the 477 patients for whom serology testing was available after the second or third dose, the global seropositivity rate was 69% (good responders: 46.3%). Immunosuppressive drugs, graft function, age, interval from transplantation, body mass index, and sex were associated with vaccine response. The devised score was strongly associated with the seropositivity rate (AUC = 0.752, *p* < 0.0001) and the occurrence of a good antibody response (AUC = 0.785, *p* < 0.0001). Notably, antibody titers declined over time both after the second and third vaccine doses. In summary, a high burden of comorbidities and immunosuppression was correlated with a weaker antibody response. A fourth vaccine dose and/or pre-exposure prophylaxis with monoclonal antibodies should be considered for KTRs who remain unprotected.

## 1. Introduction

A history of kidney transplantation confers a higher risk of COVID-19-related complications and poor outcomes, including hospitalization, intensive care unit admission, and mortality [1,2]. While kidney transplant recipients (KTRs) have been prioritized for anti-SARS-CoV-2 vaccination, immunocompromised patients were not included in pivotal efficacy trials of mRNA-based COVID-19 vaccines [3]. In addition, primary vaccination campaigns in KTRs were started with limited experimental data on their efficacy and potential complications on immunocompromised hosts. Several studies have shown that the magnitude of antibody response after COVID-19 vaccination may be limited in KTRs [4,5,6,7] and severe SARS-CoV-2 infections have been reported in vaccinated KTRs [8] While a third dose of an mRNA vaccine may be useful in inducing a boost in immune response, some patients are still unable to achieve a sufficient protection. Importantly, KTRs display marked inter-individual variations in the magnitude and occurrence of immune responses to mRNA-based vaccines, with some being able to obtain acceptable humoral and cell-mediated protection after two doses and others unable to achieve adequate responsiveness even after three doses [9,10,11] The aim of this large single-center observational study was to identify the predictive factors for serological response to the mRNA-1273 (Spikevax; Moderna) vaccine in KTRs. We also devised a score to optimize prediction of vaccine response and developed a personalized vaccination strategy based on the initial humoral response.

## 2. Patients and Methods

### 2.1. Study Population

The study population consisted of KTRs followed on an outpatient basis in the Department of Kidney Transplantation of the Strasbourg University Hospital (France). All of the KTRs included in the current investigation had received at least two doses of the mRNA-1273 vaccine between 16 February 2021 and 22 April 2021. Exclusion criteria were as follows: incomplete follow-up data, including missing SARS-CoV-2 serology after the second dose; a history of COVID-19 before the first dose; and a positive anti-SARS-CoV-2 pre-vaccination serology. Clinical and laboratory data were retrieved from digitalized medical records and the following variables were collected: patient characteristics, presence of comorbidities, transplantation-related data, and immunosuppressive drugs (including trough levels and dosages). The occurrence and severity of symptomatic COVID-19 after vaccination were carefully recorded. The study was registered at Clinicaltrials.gov (registration number: NCT04828460) and received ethical approval from the local Institutional Review Board (approval number: CE-2021-9).

### 2.2. Monitoring of Serological Response and Vaccination Strategy

Anti-Receptor Binding Domain (RBD) IgG titers were quantified using the ARCHITECT IgG II Quant test (Abbott, Abbott Park, IL, USA; detection range: 1–11,267 BAU/mL; positive agreement: 99.4%; negative agreement: 99.6%). Titers where subsequently converted according to the WHO standard and expressed in BAU/mL. The results of this assay correlate with the in vitro neutralization of ancestral SARS-CoV-2. According to the manufacturer’s instructions, patients with titers ≥7.1 and <7.1 BAU/mL were considered as seropositive and seronegative, respectively. In the former group, those who harbored titers < 143 BAU/mL were considered as moderate responders; conversely, those with titers ≥ 143 BAU/mL were classified as good responders. This threshold was found to correlate with the presence of neutralizing antibodies against ancestral strains and a vaccine efficacy >70% [12] In the latter group, those who harbored titers < 1 BAU/mL were considered as non-responders; conversely, those with titers between 1 and 7.1 BAU/mL were classified as weak responders. Anti-RBD IgG titers were measured at the following time points: (1) prior to the first vaccine dose, (2) one month after each vaccine dose, and (3) three and six months after the last vaccine dose. Patients who showed an antibody titer below 143 BAU/mL after the second vaccine dose (i.e., those classified as non-, weak, and moderate responders) were eligible to receive a third dose according to the initial vaccine scheme.

### 2.3. Statistical Analysis

Continuous data are summarized as means and standard deviations and their distribution was analyzed using the unpaired and paired Student’s *t*-tests, as appropriate. Categorical variables are expressed as counts and percentages and the χ^2^ test was used for comparison. Receiver operating characteristic (ROC) curves were generated to investigate the relationships between continuous variables and response to vaccination. To identify the independent predictors of antibody response after the second and third vaccine doses, univariate and multivariate logistic regression models were constructed. Factors associated with response to the second vaccine dose vaccine response were used to devise an empirical scale for the prediction of serological response (including both seropositivity and good response). The coefficient for each variable was inversed (1 divided by odds ratio [OR]), with the final score calculated as the sum of coefficients for each parameter. Data were analyzed using GraphPad Prism 9.0 (GraphPad Software Inc., San Diego, CA, USA), with a significance level set at *p* < 0.05 (two-tailed).

## 3. Results

### 3.1. Patients Characteristics and Serological Response after the First Vaccine Dose

We initially identified a total of 734 KTRs who had received at least one mRNA-1273 vaccine dose. After exclusion of patients with a previous history of COVID-19 or SARS-CoV-2 seropositivity (*n* = 45) and those with incomplete serological follow-up data (*n* = 123), the study sample consisted of 566 KTRs (Figure 1). Two patients developed symptomatic COVID-19 at day 17 (D17) and D24 after the first vaccine dose, respectively and two others developed COVID-19 after the second dose but before the serological assessment. The remaining 562 patients (mean age: 56.7 ± 13.1 years) were predominantly men (61.2%). The mean time elapsed from transplantation was 9.4 ± 6.4 years. Triple-drug immunosuppression with a combination of tacrolimus, mycophenolate mofetil (MMF)/mycophenolic acid (MPA), and steroids was being administered to 34.5% of participants. The general characteristics of the study patients are summarized in Table 1. One month after the first vaccine dose, the mean antibody titer was 20.7 ± 218.9 BAU/mL. The observed rate of SARS-CoV-2 seropositivity was as low as 11.7% (*n* = 66/562, Figure 2).

### 3.2. Serological Response after the Second Vaccine Dose

One month after the second vaccine dose, the seropositivity rate increased to 46.6% (*n* = 262); notably, 125 patients (22.2%) were classified as good responders (Figure 2). The characteristics of patients according to their immune response are presented in Table 1 and Appendix A. The mean antibody titers increased significantly from 20.7 ± 218.9 BAU/mL to 296.7 ± 1011 BAU/mL after the second dose (*p* < 0.0001, Figure 3). Titers were significantly higher in patients who had already achieved seropositivity after the first dose compared with those who did not (1939 ± 2265.3 BAU/mL vs. 79.6 ± 268.7 BAU/mL, respectively, *p* < 0.0001). Multivariate analysis (Table 2) identified the following variables as independently associated with seroconversion after the second dose: treatment with MMF/MPA (OR = 0.29, 95% confidence interval [CI] = 0.15−0.56, *p* = 0.0002), treatment with tacrolimus (OR = 0.54; 95% CI = 0.35−0.83, *p* = 0.005), treatment with belatacept (OR = 0.18, 95% CI = 0.04−0.86, *p* = 0.03), treatment with steroids (OR = 0.58, 95% CI = 0.38−0.88, *p* = 0.01), creatinine (for each 100 µmol/L increase; OR = 0.37, 95% CI = 0.25−0.55, *p* < 0.0001), time from transplantation (for each 10-year decrease; OR = 0.57, 95% CI = 0.42−0.77, *p* = 0.0002), and age (for each 10-year increase; OR = 0.82, 95% CI = 0.69−0.97, *p* = 0.02). We also found that tacrolimus and cyclosporine trough levels, as well as MMF/MPA dosages, were lower in patients who achieved seroconversion (Figure 4). Variables associated with vaccine response were used to devise an empirical scale for predicting seroconversion (Table 3). The final score showed a strong association with both seropositivity (AUC = 0.752, *p* < 0.0001, Figure 5A) and good response to vaccination (AUC = 0.785, *p* < 0.0001, Figure 5B). Patients with a score of 15 or higher were seronegative in 76% of cases.

Thirteen patients developed symptomatic COVID-19 after the second vaccine dose (median interval from vaccination to disease onset: 34 days; range: 4−86 days). Assessment of serological response in a subgroup of eleven patients revealed that six were non-responders, four weak responders, and one a moderate responder (antibody titer: 99.7 BAU/mL). Of the 13 patients, five required hospitalization and four presented with severe disease (oxygen requirements > 6 L/min).

#### Examples

60-year old patient, 2 years post-transplantation, under tacrolimus/MMF/steroids, serum creatinine 150 µmol/L.
7.2 − 0.4 + 1.9 + 3.4 + 1.7 + 4.1 = 18

20-year old, 7 years post-transplantation, under belatacept/MMF/steroids, serum creatinine 200 µmol/L.
2.4 − 0.8 + 5.6 + 3.4 + 1.7 + 5.4 = 17.7

35-year old, 11 years post-transplantation, under Cyclosporinee/everolimus, serum creatinine 80 µmol/L.
3.6 − 1.9 + 0 + 0 + 2.2 = 3.9

### 3.3. Serological Response after the Third Vaccine Dose

Of the 562 KTRs included in the study, 349 received a third vaccine dose and 309 underwent assessment of serological response. The characteristics of patients according to their immune response are summarized in Table 4 and Appendix A. Before the third dose, 140 (45.3%) KTRs were classified as non-responders, 90 (29.1%) as weak responders, 73 (23.6%) as moderate responders, and 5 (1.6%) as good responders. After the third dose, the seropositivity rate increased to 61.2% (from 78 to 189 patients); notably, 101 patients (32.7%) were classified as good responders. The mean antibody titers increased significantly from 142 ± 637 BAU/mL to 598 ± 864 BAU/mL (*p* < 0.0001, Figure 6). Multivariate analysis (Table 5) identified the following variables as independently associated with a lower likelihood of seroconversion after the third dose: treatment with tacrolimus (OR = 0.44, 95% CI = 0.22−0.85, *p* = 0.02), treatment with belatacept (OR = 0.03, 95% CI = 0.003−0.32, *p* = 0.003), treatment with steroids (OR = 0.4, 95% CI = 0.21−0.77; *p* = 0.006), creatinine (for each 100 µmol/L increase; OR = 0.42, 95% CI = 0.23−0.72; *p* = 0.002), time from transplantation (for each 10-year decrease; OR = 0.38, 95% CI = 0.23−0.62; *p* = 0.0002), and age (for each 10-year increase; OR = 0.62, 95% CI = 0.47−0.81; *p* = 0.0004). Conversely, a higher body mass index (for each 1 m/kg^2^; OR = 1.06, 95% CI = 1.01−1.12, *p* = 0.01) and male sex (OR = 3.07 95% CI = 1.7−5.5, *p* = 0.0002) were associated with a higher probability of seroconversion. Neither doses nor trough concentrations of immunosuppressive drugs were associated with the serological response (Figure 7).

Four patients developed symptomatic COVID-19 after the third vaccine dose (interval from vaccination to disease onset: 45, 101, and 113 days, respectively; the interval was unknown for the remaining patient). Of them, three required hospitalization and two were admitted to an intensive care unit; one ultimately died of disease. The serological response was available for three of the four patients; the results revealed that one was a non-responder (antibody titer: 1 BAU/mL) and the other two a moderate responder (antibody titer: 40.4 BAU/mL and 8 BAU/mL).

### 3.4. Overall Response in the Entire Cohort

Of the 477 patients who had serology data after the last vaccine dose, the seropositivity rate was 69% (*n* = 328); of them, 221 (46.3%) were good responders.

### 3.5. Six-Month Antibody Kinetics after the Second Vaccine Dose

The kinetics of anti-RBD antibodies was assessed in 100 KTRs after a mean interval of 97 ± 15 days from the second vaccine dose (M3). From M1 to M3, mean antibody titers decreased significantly from 992.8 ± 1546 BAU/mL to 499.7 ± 788.6 BAU/mL (*p* < 0.0001, Figure 8a). The number of good responders at M1 and M3 was 72 and 62, respectively. In 38 patients, mean antibody titers were also assessed after a mean interval of 183.6 ± 16.4 days from the second vaccine dose (M6). From M3 to M6, mean antibody titers decreased significantly from 754.6 ± 673.3 BAU/mL to 263.1 ± 598.1 BAU/mL (*p* = 0.01, Figure 8a). With the exception of one patient, all KTRs were good responders at both M3 and M6.

### 3.6. Three-Month Antibody Kinetics after the Third Vaccine Dose

The kinetics of anti-RBD antibodies were assessed in 136 KTRs after a mean interval of 109 ± 18 days from the third vaccine dose (M3). From M1 to M3, mean antibody titers decreased significantly from 401.7 ± 860.2 BAU/mL to 296.1 ± 588 BAU/mL (*p* = 0.01, Figure 8b). The number of good responders at M1 and M3 was 48 and 45, respectively.

## 4. Discussion

In the present study, we examined the humoral response elicited by two and three doses of an anti-SARS-CoV-2 mRNA-based vaccine in a large single-center cohort of KTRs. Both the seroconversion rate and the magnitude of the antibody response were taken into account. We also investigated the main predictors of humoral response and the antibody kinetics over time.

Of the 562 KTRs who were monitored after two vaccine doses, the seropositivity rate was as low as 46.6% and the antibody response was generally weak. While immunosuppressive therapy with belatacept and antimetabolites was independently associated with a less pronounced antibody response, tacrolimus and steroids may have produced synergistic effects. Moreover, a lower immune response was associated with elevated through levels of calcineurin inhibitors and high MMF/MPA doses, as well as older age and a reduced kidney function. These findings are in keeping with published data showing that the seroconversion rates of solid organ transplant (SOT) recipients who had received two mRNA-based vaccine doses ranged between 30% and 48% [6,7,13,14] However, a seronversion rate as low as 6% has been described for SOT recipients being treated with belatacept [15,16]. The use of antimetabolites has been shown to reduce the humoral response to vaccination in a dose-dependent fashion both in SOT and non-SOT recipients [7,17,18]. However, the potential impact of CNI through levels on antibody response remains controversial [7,19].

In this study, we found that a third vaccine dose given to KTRs who were originally classified as non- or weak responders increased the seroconversion rate to 69% in the entire cohort. Previous investigations conducted in SOT patients have administered the third dose without specifically taking into account the response to the previous two doses. Their findings revealed that the booster elicited enhanced T cell and neutralizing antibody responses [10,11,20,21]. The observed seroconversion rates (62−69%) are in accordance with those reported in our study [11,20,21]. We also found that a lower antibody response was independently associated with the use of immunosuppressive drugs (i.e., belatacept and tacrolimus), kidney dysfunction, a longer interval from transplantation, female sex, and age. Conversely, MMF/MPA doses and CNI through levels were not. These data are generally consistent with those reported in the literature. Masset et al. [20] have previously shown a lower antibody response in female recipients as well as in patients with kidney dysfunction and lymphopenia. Another study found that age, kidney dysfunction, MPA, belatacept, and a low lymphocyte count were associated with an impaired response to vaccination [11].

In addition to seroconversion rate, we also focused on the magnitude of antibody response. This was motivated by the observation that COVID-19 occurred even in some KTRs who successfully seroconverted. Patients with titers ≥ 143 BAU/mL were considered as good responders because this cutoff has been shown to correlate with the appearance of neutralizing antibodies against the main variants of concerns at the time of study [22] and has been associated with a vaccine protection >70% [12,23] While the frequency of good responders increased from 22.2% to 46.5% after the third dose, more than 50% of the study participants still failed to achieve adequate protection. The increase rate of good responders after each vaccine dose and the satisfactory response observed in KTRs with a previous history of COVID-19 (even after a single administration) suggested a significant effect of the antigen dose [24]. Several strategies have been proposed to increase the effectiveness of anti-SARS-CoV-2 vaccines among people who are immunocompromised. In this regard, a fourth dose can improve immunogenicity especially in the subset of patients who experienced a weak response after the third dose [25] Another strategy to provide protection is the use of anti-SARS-CoV-2 monoclonal antibodies. Transient immunosuppression reduction may also improve the immune response to vaccination, albeit at the expense of an increased rejection risk. In a small cohort of 29 KTRs, a fourth dose of SARS-CoV-2 vaccine administered during MMF/MPA hold was associated with a seroconversion rate as high as 76% [26] A randomized clinical trial to evaluate this strategy is currently ongoing [27] In patients with rheumatic and musculoskeletal diseases, a temporary hold of MMF increased the humoral response to SARS-CoV-2 vaccination from 65% to 92% [18].

In the current study, we were able to devise a scoring system to predict the response of KTRs after two vaccine doses. When the score was >15, 76% of our patients were unable to mount an adequate humoral protection. Thus, this scale can be used to identify KTRs who are in need of additional protection strategies. Antibody kinetics was assessed at M3 after the second and third doses and, in a subgroup of patients, until M6 after the second dose. The results revealed that antibody titers decreased by approximately 50% at M3, followed by another 50% decline between M3 and M6. In a study of 312 patients in whom antibody kinetics was assessed at M3 and M6 after the second dose, the seropositive rate at 6 months remained stable (61%) [28,29]. Our results on the antibody kinetics after the third dose are in line with those of Kamar et al. [30] who found decreased antibody titers at M3. However, these findings are different from those reported for KTRs with a history of COVID-19—who were able to maintain a steady antibody response at M6 [31]. The observed antibody decline in vaccinated KTRs calls to offer booster doses either according to antibody titers or at regular intervals (e.g., every six months).

One of the strengths of our study was the inclusion of a large cohort of KTRs in whom the kinetics of the antibody response was thoroughly analyzed after two and three vaccine doses. Moreover, we were able to identify several variables that were independently associated with an increased risk of inadequate humoral protection. We also correlated the occurrence of COVID-19 with the inability to mount a sufficient response to vaccination. Our findings may have relevant implications to devise individualized vaccination strategies following the second dose. However, we were unable to evaluate both antibody neutralizing activity and T-cell responses.

In conclusion, a third dose vaccine dose was found to elicit a more pronounced antibody response against SARS-CoV-2 in KTRs. While these findings suggest an antigen dose effect on immunogenicity, a satisfactory protection was achieved in less than a half of the study participants. A high burden of comorbidities and immunosuppression was correlated with a weaker antibody response. A fourth vaccine dose and/or pre-exposure prophylaxis with monoclonal antibodies should be considered for KTRs who remain unprotected after three doses. Since protective antibodies decreased over time, further studies are necessary to investigate the benefits and timing of booster doses.

## Figures and Tables

**Figure 1 jpm-12-01107-f001:**
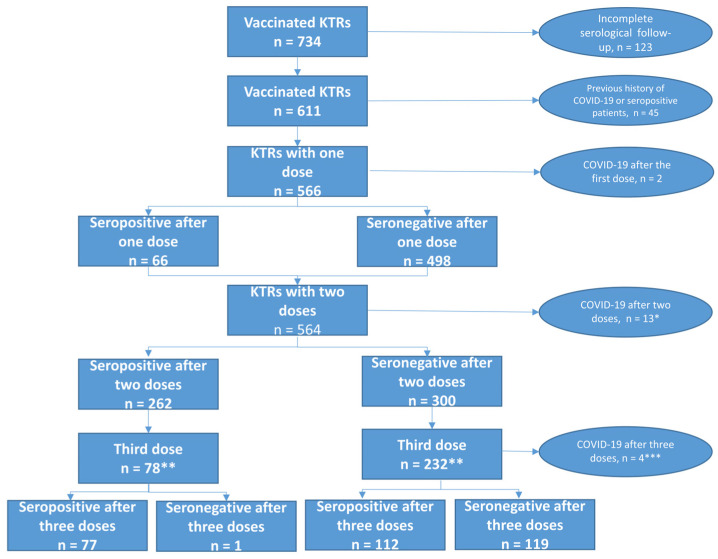
Flow of patients through the study according to the serological response to the mRNA-1273 (Spikevax; Moderna) vaccine. * Among the 13 patients who developed COVID-19 after the second vaccine dose, one displayed a moderate response (antibody titer: 99 BAU/mL), four were weak responders, and six non-responders. Two KTRs were diagnosed with COVID-19 before the serological assessment within one month and were not included in the analysis. ** Only patients with available serology after the third vaccine dose. *** Among the four patients with COVID-19, two displayed a moderate response and one was a non-responder one month after the third dose. The four patients developed COVID-19 in the month that followed vaccine administration; however, serology was unavailable.

**Figure 2 jpm-12-01107-f002:**
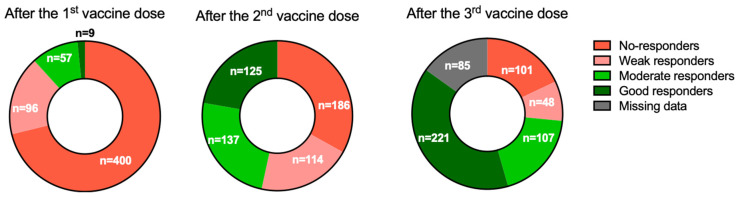
Distribution of the 562 KTRs according to the serological response assessed one month after the first, second, and third vaccine doses. Patients who harbored titers < 1 BAU/mL were considered as non-responders; conversely, those with titers between 1 and 7.1 BAU/mL were classified as weak responders. Patients with titers between 7.1 and 143 BAU/mL were considered as moderate responders; conversely, those with titers ≥ 143 BAU/mL were classified as good responders.

**Figure 3 jpm-12-01107-f003:**
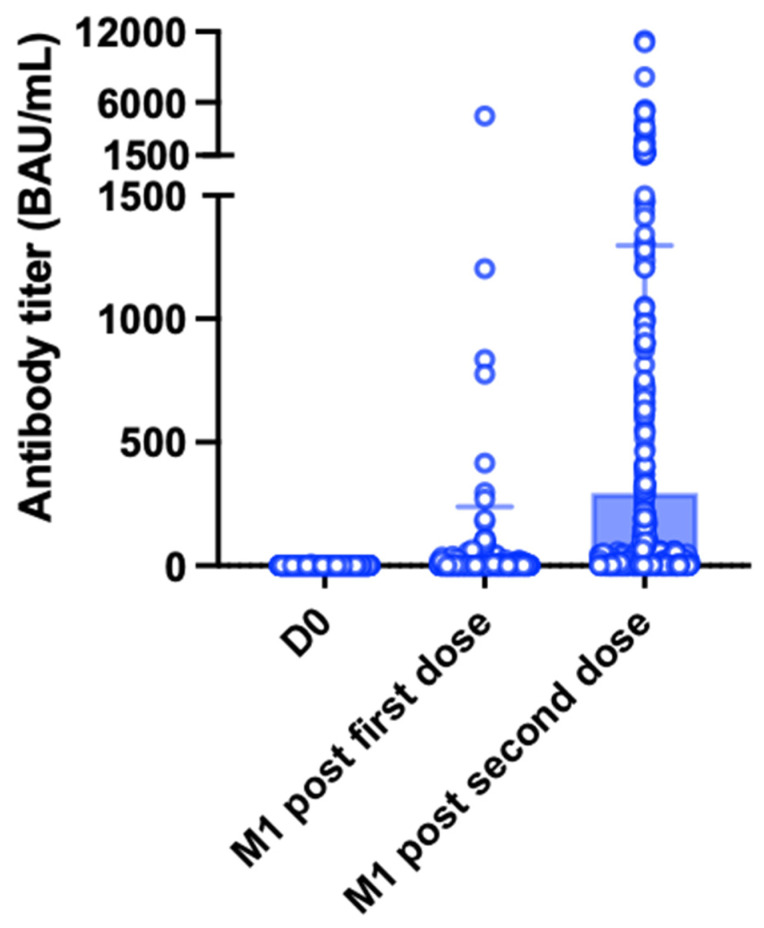
Antibody titer over time. Assessments were performed on the vaccination day (D0, *n* = 562), one month (M1) after the first dose, and one month after the second dose (*n* = 562). White dots represent individual values, whereas blue bars and the black lines indicate the means and standard deviations, respectively. After the second dose, antibody titers increased significantly from 20.7 ± 218.9 BAU/mL to 296.7 ± 1011 BAU/mL (*p* < 0.0001).

**Figure 4 jpm-12-01107-f004:**
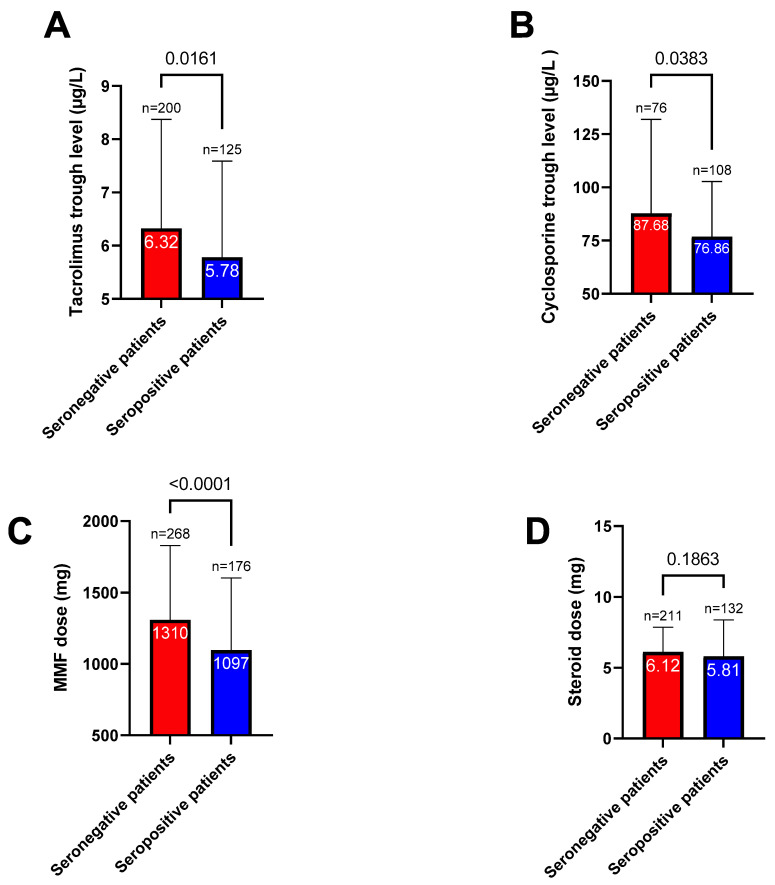
Tacrolimus trough level (**A**), Cyclosporine trough level (**B**), mycophenolate mofetil (MMF) dose (**C**), and steroid dose (**D**) according to the serological status after the second vaccine dose. Means are marked in white for each column; the number of patients and *p* values (χ^2^ test) are also reported.

**Figure 5 jpm-12-01107-f005:**
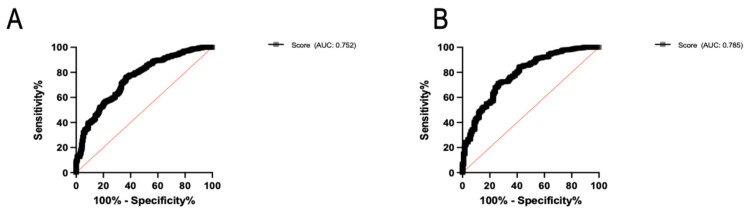
(**A**) Receiver operating characteristic (ROC) curve analysis of seroconversion (defined by an antibody titer > 7.1 BAU/mL) after two vaccine doses according to the predictive scoring system. The area under curve (AUC) for the score was 0.752 (*p* < 0.0001). (**B**) ROC curve analysis of good response (defined by an antibody titer ≥ 143 BAU/mL) after two vaccine doses according to the predictive scoring system. The area under curve (AUC) for the score was 0.785 (*p* < 0.0001).

**Figure 6 jpm-12-01107-f006:**
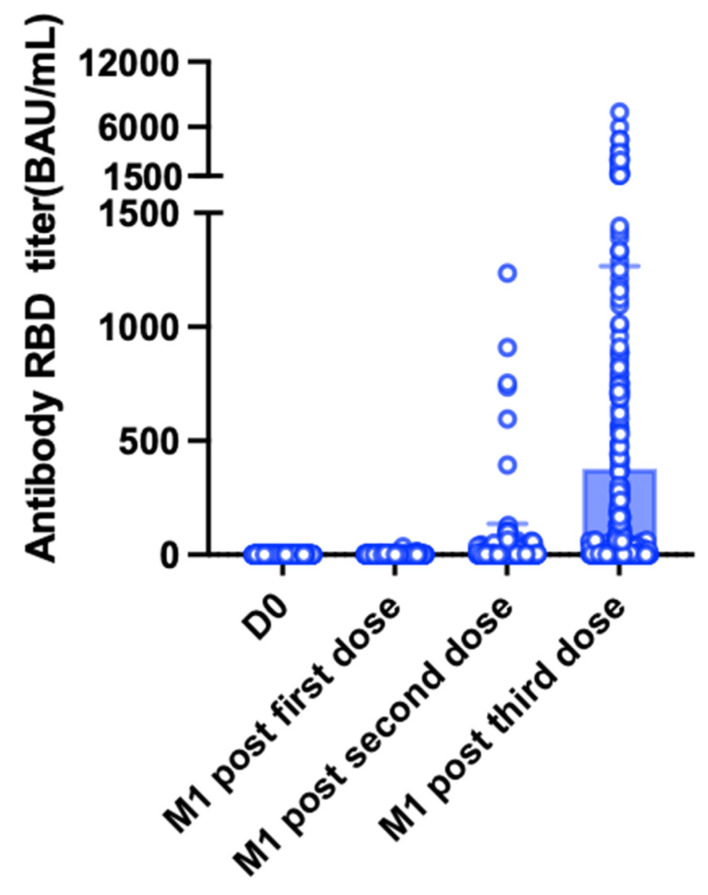
Antibody titer over time. Assessments were performed on the vaccination day (D0, *n* = 309), one month (M1) after the first dose, one month after the second dose, and one month after the third dose in the subgroup of KTRs who received three vaccine doses. White dots represent individual values, whereas blue bars and the black lines indicate the means and standard deviations, respectively. After the third dose, antibody titers increased significantly from 142 ± 637 BAU/mL to 598 ± 864 BAU/mL (*p* < 0.0001).

**Figure 7 jpm-12-01107-f007:**
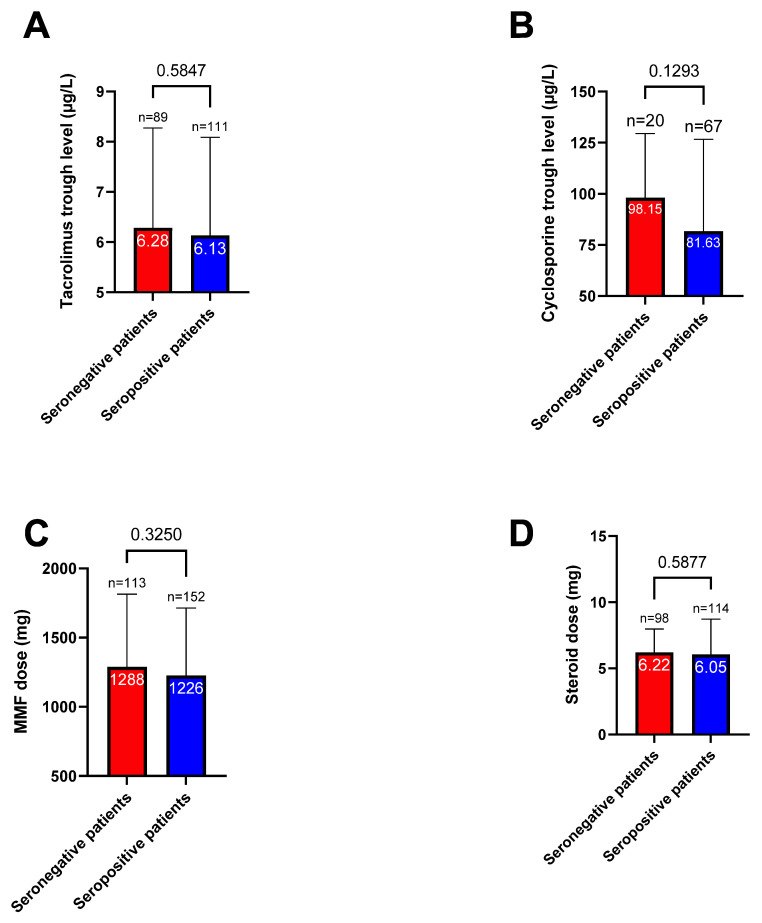
Mean tacrolimus trough level (**A**), Cyclosporine trough level (**B**), mycophenolate mofetil (MMF) dose (**C**), and steroids dose (**D**) according to the serological response after the third vaccine dose. Means are marked in white for each column; the number of patients and *p* values (χ^2^ test) are also reported.

**Figure 8 jpm-12-01107-f008:**
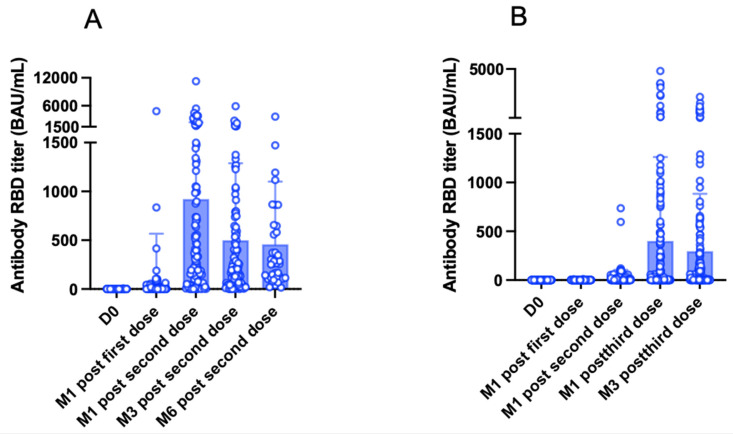
Antibody titers over time. (**A**) Assessments were performed on the vaccination day (D0), one month (M1) after the first dose, one month after the second dose, three months (M3) after the second dose (*n* = 100), and six months (M6) after the second dose (*n* = 38). White dots represent individual values, whereas blue bars and the black lines indicate the means standard deviations, respectively. In the subgroup of 100 patients, antibody titers decreased significantly from 992.8 ± 1546 BAU/mL at one month to 499.7 ± 788.6 BAU/mL at three months (*p* < 0.0001). A further decrease was observed between three and six months in the subgroup of 38 patients (from 754.6 ± 673.3 BAU/mL to 263.1 ± 598.1 BAU/mL, respectively). (**B**) Assessments were performed on the vaccination day (D0), one month (M1) after the first dose, one month (M1) after the second dose, one month (M1) after the third dose, and three months (M3) after the third dose in a subgroup of 136 patients. White dots represent individual values, whereas blue bars and the black lines indicate the means and standard deviations, respectively. Antibody titers decreased significantly from 401.7 ± 860.2 BAU/mL at M1 to 296.1 ± 588 BAU/mL at M3 (*p* = 0.01).

**Table 1 jpm-12-01107-t001:** Patient characteristics according to the absence or presence of seroconversion after two vaccine doses.

	Entire Cohort (*n* = 562)	SARS-CoV-2 Seronegative Patients (*n* = 300)	SARS-CoV-2 Seropositive Patients (*n* = 262)	*p*
Age (years)	56.7 (±13.1)	57.3 (±13)	56 (±12.3)	0.25
Male sex	344 (61.2%)	180 (60%)	164 (62.6%)	0.53
Comorbidities				
BMI (kg/m^2^)	26.6 (±5.7)	26.7 (±5.8)	26.6 (±5.5)	0.86
Cardiovascular disease	148 (26.34%)	80 (26.7%)	68 (26%)	0.85
Diabetes	204 (36.3%)	127 (42.3%)	77 (29.4%)	0.001
History of cancer	98 (17.4%)	57 (19%)	41 (15.7%)	0.3
Hypertension	472 (84%)	254 (84.7%)	218 (83.2%)	0.64
Time from kidney transplantation (years)	9.4 (±6.4)	7.5 (±6.9)	11.6 (±8.6)	<0.0001
First transplantation	474 (84.3%)	243 (81%)	231 (88.2%)	0.02
Living donor	106 (18.9%)	63 (21%)	43 (16.4%)	0.17
CNI				<0.0001
Tacrolimus	328 (58.4%)	201 (67%)	127 (48.5%)	
Cyclosporine	184 (32.7%)	76 (25.3%)	108 (41.2%)	
No CNI	50 (8.9%)	23 (7.7%)	27 (10.3%)	
Tacrolimus dose (mg) *	5.4 (±3.1)	5.8 (±3.4)	4.8 (±2.6)	0.03
Tacrolimus trough level (µg/L) **	6.1 (±2)	6.3 (±2.1)	5.8 (±1.8)	0.02
Cyclosporine dose (mg)	151.1 (±55.7)	155.8 (±65.6)	147.8 (±47.6)	0.33
Cyclosporine trough level (µg/L)	81.3 (±35)	87.7 (±44.3)	76.9 (±25.9)	0.04
MMF/MPA	448 (79.7%)	269 (89.7%)	179 (68.3%)	<0.0001
MMF/MPA dose (mg) ***	1225 (±525.3)	1310 (±521.4)	1097 (±506.3)	<0.0001
mTOR inhibitors	89 (15.8%)	27 (9%)	62 (23.6%)	<0.0001
mTOR inhibitors dose (mg) *	2.2 (±1.1)	2.1 (±0.9)	2.2 (±1.1)	0.51
mTOR inhibitors trough level (µg/L)	5.2 (±1.5)	5 (±1.5)	5.3 (±1.6)	0.38
Azathioprine	13 (2.3%)	0	13 (5%)	<0.0001
Belatacept	18 (3.2%)	16 (5.3%)	2 (0.8%)	0.002
Steroids	344 (61.2%)	212 (70.7%)	132 (50.4%)	<0.0001
Steroids dose (mg) *	6 (±2.1)	6.1 (±1.75)	5.8 (±2.6)	0.18
Tacrolimus + MMF/MPA + steroids	194 (34.5%)	142 (47.3%)	52 (19.9%)	<0.0001
Serum creatinine (µmol/L)	135.3 (±55.9)	145.2 (±58.2)	124 (±50.9)	<0.0001
Antibody titer before the second dose ****	146.4 (±1543.6)	7.5 (±3.3)	304.4 (±2249)	0.02

Data are expressed as means (±standard deviations) or counts (percentages), as appropriate. Abbreviations: BMI, body mass index; CNI, calcineurin inhibitors; MMF, mycophenolate mofetil; MPA, mycophenolic acid; mTOR, mammalian target of rapamycin. * Missing data for one patient, ** Missing data for three patients, *** Missing data for four patients, **** Missing data for two patients.

**Table 2 jpm-12-01107-t002:** Univariate and multivariate analysis of predictors factors for seroconversion after two vaccine doses.

	Univariate Odds Ratio (95% Confidence Interval)	*p*	Multivariate Odds Ratio (95% Confidence Interval)	*p*
Age (for each 10-year increase)	0.93 (0.81–1.05)	0.2	0.82 (0.69–0.97)	0.02
Male sex	1.12 (0.79–1.57)	0.53		
Comorbidities				
BMI (for each 1 kg/m^2^)	0.99 (0.97–1)	0.87		
Cardiovascular disease	1.0 (0.66–1.41)	0.85		
Diabetes	0.57 (0.4–0.81)	0.002	0.70 (0.45–1.06)	0.09
Cancer	0.79 (0.51–1.23)	0.29		
Hypertension	0.9 (0.57–1.41)	0.63		
Time from transplantation (for each 10-year decrease)	0.49 (0.38–0.61)	<0.0001	0.57 (0.42–0.77)	0.0002
First transplantation	1.75 (1.1–2.8)	0.02	1.15 (0.66–2.02)	0.63
Living donor	0.74 (0.48–1.13)	0.17	0.87 (0.53–1.44)	0.60
TAC vs. CS or no CNI	0.46 (0.33–0.65)	<0.0001	0.54 (0.35–0.83)	0.005
MMF/MPA	0.25 (0.16–0.39)	<0.0001	0.29 (0.15–0.56)	0.0002
mTOR inhibitors	3.13 (1.93–5.1)	0.003	1.64 (0.82–3.27)	0.17
Azathioprine	32.5 (1.73–611.5)	0.02		
Belatacept	0.14 (0.03–0.60)	0.008	0.18 (0.04–0.86)	0.03
Steroids	0.42 (0.3–0.6)	0.01	0.58 (0.38–0.88)	0.01
Serum creatinine (for each 100 µmol/L increase)	0.46 (0.31–0.65)	<0.0001	0.37 (0.25–0.55)	<0.0001

Abbreviations: BMI, body mass index; TAC, tacrolimus; CS, cyclosporine; CNI, calcineurin inhibitors; MMF, mycophenolate mofetil; MPA, mycophenolic acid; mTOR, mammalian target of rapamycin.

**Table 3 jpm-12-01107-t003:** Variables included in the score for predicting serological response to vaccination.

Variable	Coefficient
Belatacept	5.6
Tacrolimus	1.9
MMF/MPA	3.4
Steroids	1.7
Serum creatinine (for each 100 µmol/L increase)	2.7
Age (for each 10-year increase)	1.2
Time from transplantation (for each 10-year decrease)	−1.75

Abbreviations: MMF, mycophenolate mofetil; MPA, mycophenolic acid.

**Table 4 jpm-12-01107-t004:** Patient characteristics according to the absence or presence of seroconversion after three vaccine doses.

	Entire Cohort (*n* = 309)	SARS-CoV-2 Seronegative Patients (*n* = 120)	SARS-CoV-2 Seropositive Patients (*n* = 189)	*p*
Age (years)	57.2 (±12.5)	59.1 (±13)	55.9 (±12.1)	0.02
Male sex	196 (63.4%)	62 (51.7%)	134 (70.9%)	0.0006
Comorbidities				
BMI (kg/m^2^)	26.7 (±6)	26.1 (±5.6)	27 (±6.3)	0.17
Cardiovascular disease	77 (24.9%)	35 (29.2%)	42 (22.2%)	0.17
Diabetes	124 (40.1%)	52 (43.3%)	72 (38.1%)	0.36
Cancer	61 (19.7%)	28 (23.3%)	33 (17.4%)	0.21
Hypertension	265 (85.8%)	104 (86.7%)	161 (85.2%)	0.72
Time from kidney transplantation (years)	7.9 (±7.1)	6 (±6.2)	9.2 (±7.4)	<0.0001
First transplantation	250 (80.9%)	94 (78.3%)	156 (82.5%)	0.36
Living donor	63 (20.4%)	20 (16.7%)	43 (22.8%)	0.2
CNI				0.001
Tacrolimus	201 (65.1%)	89 (74.2%)	112 (59.3%)	
Cyclosporine	87 (28.2%)	20 (16.7%)	67 (35.5%)	
No CNI	21 (6.8%)	11 (9.2%)	10 (5.3%)	
Tacrolimus dose (mg) *	5.6 (±3.3)	6.2 (±3.6)	5.1 (±2.9)	0.02
Tacrolimus trough level (µg/L) *	6.2 (±2)	6.3 (±2)	6.1 (±2)	0.5
Cyclosporine dose (mg)	160 (±57)	185 (±56)	153 (±56)	0.03
Cyclosporine trough level (µg/L)	85.4 (±42.7)	98.1 (±31.3)	81.6 (±45)	0.07
MMF/MPA	266 (86.1%)	113 (94.2%)	153 (81%)	0.001
MMF/MPAs dose (mg) *	1253 (±507)	1290 (±529)	1225 (±490)	0.3
mTOR inhibitors	34 (11%)	6 (5%)	28 (14.8%)	0.01
mTOR inhibitors dose (mg)	2 (±0.9)	1.8 (±0.7)	2 (±0.9)	0.61
mTOR inhibitors trough level (µg/L)	5 (±1.6)	4.9 (±1.4)	5.1 (±1.6)	0.82
Azathioprine	3 (1%)	0	3 (1.6%)	0.17
Belatacept	12 (3.9%)	11 (9.2%)	1 (0.5%)	0.0004
Steroids	213 (68.9%)	99 (82.5%)	114 (60.3%)	<0.0001
Steroids dose (mg) *	6.1 (±2.3)	6.2 (±1.8)	6.1 (±2.7)	0.58
Tacrolimus + MMF/MPA + steroids	137 (44.4%)	73 (60.8%)	64 (33.9%)	<0.0001
Serum creatinine (µmol/L)	139 (±54.2)	150.3 (±62.6)	131.8 (±46.9)	0.003
Antibody titer before the third dose	142 (±637)	10.5 (±23.5)	226 (±804)	0.0003

Data are expressed as means (±standard deviations) or counts (percentages), as appropriate. Abbreviations: BMI, body mass index; CNI, calcineurin inhibitors; MMF, mycophenolate mofetil; MPA, mycophenolic acid; mTOR, mammalian target of rapamycin. * Missing data for one patient.

**Table 5 jpm-12-01107-t005:** Univariate and multivariate analysis of predictors factors for seroconversion after three vaccine doses.

	Univariate Odds Ratio (95% Confidence Interval)	*p*	Multivariate Odds Ratio (95% Confidence Interval)	*p*
Age (for each 10-year increase)	0.81 (0.67–0.98)	0.03	0.62 (0.47–0.81)	0.0004
Male sex	2.28 (1.42–3.68)	0.0007	3.07 (1.7–5.5)	0.0002
Comorbidities				
BMI (for each 1 kg/m^2^)	1.03 (0.99–1.07)	0.18	1.06 (1.01–1.12)	0.01
Cardiovascular disease	0.69 (0.41–1.17)	0.17	1.31 (0.68–2.52)	0.42
Diabetes	0.81 (0.51–1.28)	0.36		
Cancer	0.67 (0.4–1.23)	0.21		
Hypertension	0.89 (0.45–1.70)	0.72		
Time from transplantation (for each 10-year decrease)	0.46 (0.30–0.67)	0.0002	0.38 (0.23–0.62)	0.0002
First transplantation	1.31 (0.73–2.32)	0.36		
Living donor	1.47 (0.83–2.7)	0.2	1.49 (0.72–3.05)	0.28
TAC vs. CS or no CNI	0.51 (0.30- 0.83)	0.0078	0.44 (0.22–0.85)	0.02
MMF/MPA	0.26 (0.10–0.58)	0.002	0.27 (0.06–1.16)	0.08
mTOR inhibitors	3.30 (1.41–9.07)	0.01	1.4 (0.3–6.78)	0.68
Belatacept	0.05 (0.003–0.28)	0.005	0.03 (0.003–0.32)	0.003
Steroids	0.32 (0.18–0.55)	<0.0001	0.4 (0.21–0.77)	0.006
Serum creatinine (for each 100 µmol/L increase)	0.53 (0.33–0.81)	0.005	0.42 (0.23–0.72)	0.002

Abbreviations: BMI, body mass index; TAC, tacrolimus; CS, cyclosporine; CNI, calcineurin inhibitors; MMF, mycophenolate mofetil; MPA, mycophenolic acid; mTOR, mammalian target of rapamycin.

## Data Availability

The data that support the findings of this study are available from the corresponding author upon reasonable request.

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
