# Peer review of "Prediction of Vaccine Response and Development of a Personalized Anti-SARS-CoV-2 Vaccination Strategy in Kidney Transplant Recipients: Results from a Large Single-Center Study"

_jpm, 2022, doi:10.3390/jpm12071107_

Round 1

Reviewer 1 Report

All in all acceptable 

Author Response

Response: Thank you for your valuable comments. 

Reviewer 2 Report

This is a very relevant subject and pertinent to clinical practice. I read this paper with great enthusiasm.  

Author Response

This is a very relevant subject and pertinent to clinical practice. I read this paper with great enthusiasm.  

Response: We greatly appreciate the reviewer’s positive feedback.

Reviewer 3 Report

This is a nice description of vaccination in transplant recipients.  The manuscript itself is well written.  The abstract contains a couple of errors, and the authors should go over it line by line to correct the minor mistakes.  This should be a nice contribution.

Author Response

AUTHORS’ RESPONSE TO COMMENT FROM Reviewer #3

This is a nice description of vaccination in transplant recipients.  The manuscript itself is well written. 

Response: We are very grateful for your supportive comments and valuable contribution.

The abstract contains a couple of errors, and the authors should go over it line by line to correct the minor mistakes.  This should be a nice contribution.

Response: Thank you for highlighting this. This has now been amended